# Biomechanical Difference between Conventional Transtibial Single-Bundle and Anatomical Transportal Double-Bundle Anterior Cruciate Ligament Reconstruction Using Three-Dimensional Finite Element Model Analysis

**DOI:** 10.3390/jcm10081625

**Published:** 2021-04-12

**Authors:** Jae Gyoon Kim, Kyoung Tak Kang, Joon Ho Wang

**Affiliations:** 1Department of Orthopedic Surgery, Ansan Hospital, Korea University College of Medicine, Ansan-si 15355, Gyeonggi-do, Korea; jgkim7437@gmail.com; 2Department of Mechanical Engineering, Yonsei University, Seoul 03722, Korea; tagi1024@yonsei.ac.kr; 3Department of Orthopedic Surgery, Samsung Medical Center, Sungkyunkwan University School of Medicine, Seoul 06351, Korea

**Keywords:** knee, anterior cruciate ligament, double bundle, single bundle, finite element model, graft stress, contact stress

## Abstract

The purpose of our study was to analyze the graft contact stress at the tunnel after transtibial single-bundle (SB) and transportal double-bundle (DB) anterior cruciate ligament (ACL) reconstruction. After transtibial SB (20 cases) and transportal DB (29 cases) ACL reconstruction, the three-dimensional image of each patient made by postoperative computed tomography was adjusted to the validation model of a normal knee and simulated SB and DB ACL reconstructions were created based on the average tunnel position and direction of each group. We also measured graft and contact stresses at the tunnel after a 134 N anterior load from 0° to 90° flexion. The graft and contact stresses became the greatest at 30° and 0° flexion, respectively. The total graft and contact stresses after DB ACL reconstruction were greater than those after SB ACL reconstruction from 0° to 30° and 0° to 90° knee flexion, respectively. However, the graft and contact stresses of each graft after DB ACL reconstruction were less than those after SB ACL reconstruction. In conclusion, the total graft and total contact stresses after DB ACL reconstruction are higher than those after SB ACL reconstruction from 0° to 30° and 0° to 90° knee flexion, respectively. However, the stresses of each graft after DB ACL reconstruction are about half of those after SB ACL reconstruction.

## 1. Introduction

Recently, placing a graft within the anterior cruciate ligament (ACL) footprint has been emphasized in anatomical ACL reconstruction [1]. Anatomical ACL reconstruction means that tunnels are made in the femoral and tibial ACL footprints regardless of the number of bundles [2]. In conventional single-bundle (SB) ACL reconstruction, the tunnels are made in the posterolateral (PL) tibial footprint and the anteromedial (AM) femoral footprint, resulting in a non-anatomical and more vertical direction than the native ACL, which cannot restore rotatory laxity [3,4]. Anatomical double-bundle (DB) ACL reconstruction shows superior biomechanical results, including both anterior and rotatory stability, compared to conventional non-anatomical SB ACL reconstruction [5,6,7,8]. 

Better positioning of a femoral tunnel anatomically would be accomplished using a femoral tunnel drilling technique independently of the tibial tunnel [9]. The necessity to make a femoral tunnel independently of a tibial tunnel has drawn interest in independent techniques such as a transportal (TP) and an outside-in technique [9,10]. We assumed that these changes in the technique of performing anatomical ACL reconstruction would also change the femoral tunnel geometry and the stress patterns of both graft and tunnel. This change in the stress pattern would affect longer-term follow-up clinical results despite the anatomical femoral tunnel position. In a previous study, the femoral graft bending angle was defined as the angle between the femoral tunnel axis and the graft, and this angle was compared between conventional SB ACL reconstruction using the transtibial (TT) technique and anatomical DB ACL reconstruction using the TP technique. The authors found that anatomical DB ACL reconstruction showed a more acute angle than conventional SB ACL reconstruction in an extended knee position, which might have increased the stress in the graft at the femoral tunnel opening [11].

Many studies have evaluated the biomechanics of the reconstructed ACL using cadaver or finite element model (FEM) analysis [12,13,14,15,16,17,18]. Some studies have compared knee kinematics and biomechanics between SB and DB ACL reconstructions [14,17]. As far as we are aware, few studies have evaluated and compared the graft stress and contact stress at the tunnel after conventional SB ACL reconstruction and anatomical DB ACL reconstruction using FEM analysis on the basis of a surgical simulation model made using computed tomography (CT) scans of patients. This type of model can represent the real tunnel position and direction after ACL reconstructions using both surgical techniques. The purpose of this FEM study was to analyze the graft stress and contact stress at the tunnel after conventional SB ACL reconstruction and anatomical DB ACL reconstruction at different angles of knee flexion. We hypothesized that the graft stress and contact stress after anatomical DB ACL reconstruction using the TP technique would be greater than those after conventional SB ACL reconstruction using the TT technique.

## 2. Materials and Methods

### 2.1. Intact Model 

A three-dimensional (3D) FEM of the lower extremity was developed based on computed tomography (CT) images. A light speed volume computed tomography (VCT, GE Medical Systems, Milwaukee, WI, USA) scanner was used. CT images of a 0.1 mm slice from a 34-year-old male subject (height 178 cm, weight 75 kg) were obtained. The 3D surface of the femur, tibia, fibula, and patella at full extension was generated by Mimics software (Materialise Inc., Leuven, Belgium) using digital CT data (Figure 1) [19].

Based on magnetic resonance imaging (MRI), the femoral cartilage, both menisci, patellar tendon, and four ligaments (anterior cruciate, posterior cruciate, medial collateral, and lateral collateral ligaments) were segmented manually in 3D reconstruction models. This segmentation was accurate to 0.1 mm. The Initial Graphics Exchange Specification (IGES) files exported from Mimics were loaded into Unigraphics (UG) NX 7.0 (Siemens PLM Software, Torrance, CA, USA) to make solid models for each femur, tibia, fibula, patella, and soft-tissue segment, which were loaded into Hypermesh 8.0 (Altair Engineering, Inc., Troy, MI, USA) to make the FE mesh (Figure 2).

The FE mesh was analyzed using ABAQUS 6.6-1 (Hibbitt, Karlsson and Sorenson, Inc., Providence, RI, USA). The cortical bone, cancellous bone, and intramedullary canal were included in the bone model. The bone parts were assumed to be rigid contrary to soft tissues according to a previous study [20]. Therefore, a primary node positioned at the center of rotation at full extension represented each bony structure. The FE models of the soft tissue comprised the meniscus and four ligaments. The meniscus and cartilage models were also developed based on a previous study [20]. The attachment between the cartilage and bones was assumed to be completely bonded. [19] The anterior and posterior horns of both menisci were attached to the tibia plateau, and the medial meniscus was additionally attached to the joint capsule along the outer edge, following Pena et al. [20]. The ligaments model was assumed to be a hyperelastic, rubber-like material, which represents the nonlinear stress–strain relations [21,22]. A strain energy potential function characterized the model [21], and the polynomial form of the strain energy potential was selected from the ABAQUS material library [23]. Soft tissues are normally exposed to in vivo residual stresses that they undergo. The initial strain model of ligaments was also made according to a previous study [20].

### 2.2. Surgical Procedure

The patients were classified into a conventional TT SB group (20 patients) and an anatomical TP DB group (29 patients). We divided the patients according to the time lapse from the injury to the reconstruction. Group 1 consisted of 20 patients who had up to a 6-month interval between the injury and reconstruction and underwent conventional SB ACL reconstruction using the TT technique, because in this case, we performed the remnant preservation technique and the quality of remnant tissue would be related with this interval. Group 2 consisted of 29 patients with more than a 6-month interval between the injury and the reconstruction and underwent anatomical DB ACL reconstruction using the TP technique with non-remnant preservation.

After the usual portal formation and arthroscopic examination, an accessory anteromedial (AAM) portal was made. The graft was made using the hamstring tendon. For SB reconstruction, four stranded grafts of semitendinosus and gracilis were made, and for DB reconstruction, six stranded grafts (triple-stranded semitendinosus for AM bundles and triple-stranded gracilis for PL bundles) were created. We harvested and used both semitendinosus and gracilis in all cases. The mean graft diameter of conventional SB ACL reconstruction was 8.1 ± 0.7 mm (range from 7 to 9 mm), and that of anatomical DB ACL reconstruction was 7.3 ± 0.8 mm (range 7 to 8 mm) for the AM graft and from 5.5 ± 0.6 mm (range 5 to 6 mm) for the PL graft.

#### 2.2.1. Conventional SB Reconstruction Using the TT Technique

To make a tibial tunnel, we used an ACL tibial guide (Linvatec, Largo, FL, USA), and the tibial guide tip was positioned on the ACL tibial footprint (PL bundle center) at the point on the extended line from the lateral meniscus anterior horn. The femoral tunnel was reamed near the AM femoral footprint at the level of the 10:30 (right knee) and 1:30 (left knee) positions, approximately 1–2 mm anterior to the posterior cortex of the femur, through the tibial tunnel previously reamed.

#### 2.2.2. Anatomical DB Reconstruction Using the TP Technique

The femoral footprints of both bundles were determined using ACL remnants and bony landmarks [23] and marked using a Steadman awl (ConMed (Linvatec), Largo, FL, USA). A 2.4 mm guide pin was advanced through the Bullseye^®^ femoral guide (ConMed (Linvatec), Largo, FL, USA) passed through the AAM portal, with the tip aimed at the center of the femoral footprints of both bundles, previously determined. The AM bundle footprint was determined 5 to 6 mm anterior to the posterior cartilage margin and 3 to 4 mm inferior to the posterolateral corner of the intercondylar notch in 90° flexion, and the center of the PL bundle was determined 5 mm superior to the margin of the articular cartilage on an imaginary line perpendicular to the tangent line of the lateral femoral condyle in 90° flexion. The anatomical center of the tibial footprint was determined using the ACL remnant. The center of the AM bundle was determined at a point posterior to the anterior ridge of ACL tibial footprint, and the center of the PL bundle was determined at a point posterior to the AM bundle footprint. A Sentinel cannulated reamer (ConMed (Linvatec), Largo, FL, USA) and a 4.5 mm EndoButton drill bit (Smith & Nephew Endoscopy, Andover, MA, USA) were then used to make femoral tunnels. Next, a tibial tunnel was reamed using an ACL tibial guide (Linvatec, Largo, FL, USA), and the guide tip was positioned at the center of the tibial footprints of both bundles.

### 2.3. Surgical Simulation Model

Three days after ACL reconstruction, CT scans were taken of all knees (49 patients) with the patients’ consent. A light speed VCT scanner (GE Medical Systems, Milwaukee, WI, USA) was used in all cases. The CT scans were taken in full extension. The collimation and tube parameters were 16 × 0.625 mm and 120 kVp/200 mA, respectively. The acquisition matrix and the field of view were 512 × 512 and 140 mm with a slice thickness of 0.625 mm, respectively. CT images of each group were used for 3D reconstruction. This 3D reconstruction model of each technique group was adjusted to the validation model. The average position of the AM and PL femoral tunnel in a parallel direction to the Blumensaat line was 23.7% ± 5.4% and 34.7% ± 6.3%, respectively, after anatomical DB ACL reconstruction. The average position of the AM and PL femoral tunnel in a vertical direction to it was 20.5% ± 6.1% and 50.1% ± 6.9%, respectively, after anatomical DB ACL reconstruction. The average position of the femoral tunnel in a parallel direction to it was 33.3% ± 3.4% and in a vertical direction to it was 40.8% ± 6.3% after conventional SB ACL reconstruction. The average AM and PL femoral graft bending angles were 111.5° ± 8.8° and 118.9° ± 9.8°, respectively, after anatomical DB ACL reconstruction. The mean femoral graft bending angle after conventional SB ACL reconstruction was 125.3° ± 11.1° [11]. To simulate conventional SB ACL reconstructions in the 0° analytic model, two 8-mm-diameter virtual tunnels were made at the average femoral and tibial tunnel positions evaluated by the methods described by Bernard et al., [24] and also made according to the direction of the tunnel estimated by the average femoral and tibial graft bending angles [11]. To simulate anatomical DB ACL reconstructions, four 6-mm-diameter virtual tunnels were made using the same method as for SB reconstruction. The simulation process was performed using UG NX 7.0. The AM and PL grafts were simulated in each tunnel with 20 N tension in an extended position. [15] The interface between the grafts and the tunnel was bonded using mesh tie kinematic constraints. The contacts between bone and ligament and between ligament and ligament were constructed using the penalty formulation, assuming 0.1 and 0.001 of frictional coefficients, respectively. [25] The validated model was combined with the actual models through surgery (Figure 3).

### 2.4. Loading and Boundary Conditions and Evaluation

The tibial translations under 0 to 100 N of the anterior and posterior forces working on the knee center at 0° extension were compared with previous studies that validated the intact knee model [26,27]. Second loading conditions included a 134 N anterior load to the tibia at 0° extension as well as at 30°, 60°, and 90° flexion. We measured and analyzed anterior tibial translation and graft stresses, which means the total stress loaded at the whole graft, and contact stresses (von-Mises stress), which means the stress loaded at the contact surface between graft and tunnel.

## 3. Results

### 3.1. Validation

The translation results for validation were similar to those of previous studies [26,27]. The tibial anterior and posterior translations for 100 N forces were 2.89 and 4.10 mm, respectively (2.43 and 5.28 mm in the experimental study and 2.55 and 4.86 mm in the computational study). The element size was decided according to a previous study [20].

### 3.2. Anterior Tibial Translation under a 134 N Anterior Load 

The anterior tibial translation (ATT) in the TT SB group and the TP DB group ranged from 4.2 mm for 0° to 9.1 mm for 60° and from 4.6 mm for 0° to 8.1 mm for 30° (Figure 4).

### 3.3. Graft Stress

The graft stress under an anterior load after ACL reconstruction using the two techniques became the greatest at 30° knee flexion (13.1 MPa for SB, 13.3 MPa for DB) (Table 1).

At lower flexion angles (0° and 30°), the total graft stress (the sum of AM and PL graft stresses) of DB ACL reconstruction was slightly higher than that of SB ACL reconstruction. However, at higher flexion angles (60° and 90°), the total graft stress in DB ACL reconstruction was lower than that in SB ACL reconstruction. It is worth noting that each AM or PL graft stress was lower than in conventional SB ACL reconstruction from 0° to 90° knee flexion. In the simulation of DB ACL reconstruction, the AM graft stress ranged from 6.4 to 4.8 MPa at 30° and 90° knee flexion, respectively, whereas PL graft stress ranged from 6.9 to 4.4 MPa at 30° and 90° knee flexion, respectively. At lower flexion angles (0° and 30°), the PL graft stress was higher, and at higher flexion angles (60° and 90°), the PL graft stress was lower than the AM graft stress (Figure 5).

### 3.4. Contact Stress Between Graft and Tunnel

The patterns of contact stress according to knee flexion are shown in Table 2 and Table 3.

In the simulation of SB ACL reconstruction, the contact stress in the femoral tunnel ranged from 4.1 MPa at 90° to 12.0 MPa at 0° flexion. In the simulation of DB ACL reconstruction, the total contact stress in the femoral tunnel ranged from 4.8 MPa at 90° to 12.5 MPa at 0° knee flexion. The total contact stress (sum of AM and PL bundle stresses) at the femoral tunnel in DB ACL reconstruction was higher than that in SB ACL reconstruction, even though each contact stress at the AM or PL femoral tunnel of DB ACL reconstruction was about half of that of SB ACL reconstruction from 0° to 90° flexion (Figure 6).

The total contact stress in the tibial tunnel ranged from 3.5 MPa at 90° to 6.0 MPa at 0° flexion in SB ACL reconstruction and 3.6 MPa at 90° to 6.2 MPa at 0° flexion in DB ACL reconstruction. The total contact stress at the tibial tunnel in DB ACL reconstruction was higher than that in SB ACL reconstruction at 0°, 60°, and 90° knee flexion, even though each contact stress at the AM or PL tibial tunnel in DB ACL reconstruction was lower than in SB ACL reconstruction from 0° to 90° flexion (Figure 7).

## 4. Discussion

The main findings of our study were that the total graft stress after anatomical DB ACL reconstruction using the TP technique is slightly higher from 0° to 30° flexion and lower from 60° to 90° flexion than SB ACL reconstruction using the TT technique. The contact stress was largest at 0° flexion and decreased from 0° to 90° flexion after ACL reconstruction using both techniques. The total contact stress at the femoral and tibial tunnels after DB ACL reconstruction was equal or higher than in SB ACL reconstruction from 0° to 90° flexion. However, the graft stress and contact stress of each AM/PL graft were about half of those in SB ACL reconstruction from 0° to 90° flexion. There have been many studies that have evaluated the biomechanics of the reconstructed ACL using cadaver or FEM analysis. [12,13,14,15,16,17,18]. However, there has not been any study in which a real patient image with the real tunnel position and direction after ACL reconstruction using both techniques was used, nor have there been any FEM studies to determine graft stress and contact stress after conventional SB and anatomical DB ACL reconstruction.

In our study, the ATT was the largest at 60° flexion after SB ACL reconstruction and at 30° flexion after DB ACL reconstruction. Our results also showed that the ATT after DB ACL reconstruction was slightly higher than that in SB ACL reconstruction at a 0° knee position with a difference of 0.4 mm and was lower from 30° to 90° knee flexion, although a maximum 1.5 mm difference in the ATT may not be clinically significant. These results were similar to another cadaver study that presented that the ATT under a 90 N anterior load after anatomical DB reconstruction was significantly less than after SB reconstruction from 30° to 90° flexion. In the same study, the maximum ATT was observed in the ACL intact knee at 30° flexion and after anatomical DB reconstruction [28]. In some other studies, DB ACL reconstructions have shown closer biomechanics to an intact knee compared with SB reconstructions [8,14,29]. Tsai et al. also showed similar result to our study in that the ATT after DB ACL reconstruction was smaller at a high flexion angle than after SB ACL reconstruction [14]. This result might be due to the anatomical tunnel positioning in DB ACL reconstruction. Yasuda et al. suggested that the tunnel location rather than the number of bundles would be the cause of better results of DB reconstruction than the conventional SB reconstruction [2].

The two bundles of the ACL function reciprocally in passive flexion and extension, with the tighter PL bundle in extension and the tighter AM bundle in flexion [30]. The ligament function evaluated in vitro might be different from that evaluated in vivo, because the physiological loading conditions would be different [31]. When an anterior load is applied to the knee, both bundles share the load in near extension, while the majority of the load is shared by the AM bundle as the knee is flexed [16,32]. Many authors have reported that AM/PL grafts are the longest in extension and decrease in flexion (from 0° to 90°) [15,33]. In many studies, ACL tension has been estimated by measuring the distance between ACL footprints [31,33,34]. However, the deformation of the graft and the impingement between the grafts and surrounding bone affect ligament tension [15,35]. Kim et al. suggested that the stress caused by graft impingement between the graft and surrounding bone according to knee flexion might maintain the tension within the grafts without actual graft lengthening [15]. Atypical materials such as ligaments lead to large amounts of stress concentration, accompanied by a relatively small load if there is any deformation or contact through translation. Therefore, the value of stress was calculated in this study. In general, the value of the cross-sectional area is required to calculate stress quantity, and thus, it is much more convenient to conduct FEM rather than cadaver experiments.

In our study, the total graft stress after DB ACL reconstruction was slightly greater at 0° to 30° flexion and lesser at 60° to 90° flexion than after SB ACL reconstruction, although there was a similar trend in both techniques. In cases of DB ACL reconstruction, an additional PL tunnel would further enhance the stress concentration around the tunnel at low knee flexion [17]. In addition, the PL graft tension maximized at a lower flexion angle and decreased with increased flexion of the knee [16]. This might be the cause of higher total graft stress after DB ACL reconstruction at a low knee flexion in our study. However, in this study, the stresses of the AM and PL grafts were about half of those in SB ACL reconstruction from 0° to 90°, although the total graft stress in DB ACL reconstruction was greater than that in SB ACL reconstruction at a low flexion angle. Yasuda et al. described that in DB reconstruction, forces loaded to the tibia are shared by the two reconstructed bundles, which can prevent excessive loading to one bundle [7]. This might be one of the advantages of DB ACL reconstruction.

In our study, the total contact stress after DB ACL reconstruction using the TP technique was greater than after SB ACL reconstruction using the TT technique at 0° to 90° knee flexion. As mentioned above, in cases of DB ACL reconstruction, additional PL tunnel creation would further amplify the severity of the stress concentration around the tunnel [17]. This might be the cause of higher total contact stress after anatomical DB ACL reconstruction from 0° to 90° flexion in our study. However, each contact stress at AM and PL femoral tunnels were less than those in SB ACL reconstruction using the TT technique. Wang et al. showed that the femoral graft bending angle of the DB ACL using the TP technique was more acute than that of the SB ACL using the TT technique and suggested that this acute bending angle might increase the graft stress at low flexion [11]. Our results presented that the graft bending stress at the femoral tunnel after applying both reconstruction techniques was greatest at 0° flexion, which we assumed in our previous study. However, our results also showed that the contact stress of the AM or PL tunnel was about half of that of the SB tunnel, regardless of the more acute graft bending angle. Therefore, if we performed DB ACL reconstruction using the TP technique, the damage to each AM or PL graft might not be more than that by SB ACL reconstruction using the TT technique. Even though the anatomical ACL reconstruction technique could increase the graft stress and contact stress at the tunnel, we would not know clearly its actual clinical effect. However, we need to study the comparison of the clinical results between anatomical ACL reconstruction and conventional ACL reconstruction to prove the clinical effect of this biomechanical difference.

The contact stress in the femoral tunnel was greater than that in the tibial tunnel in our result. Hirokawa et al. presented in a FEM study that the largest stress was observed near the femoral insertion, and the area least stretched throughout the whole range of flexion was the portion near the tibial insertion of an intact ACL [36]. Song et al. also proposed in their FEM study that the highest stress was focused near the femoral insertion site, and the least stress was shown near the tibial insertion at full extension under the anterior load [35]. Similar results were observed in our study. Therefore, graft failure might develop more often near the femoral tunnel in SB or DB ACL reconstruction.

This study had some limitations and simplifications. First, the ligament’s initial tension value was considered in only one particular case. According to findings from previous studies, the graft stress and contact stress will be different based on the initial value of ligament tension [18]. Second, we performed this FEM study under only one loading condition (134 N anterior tibial translation). If we combined various loading condition such as varus/valgus and internal/external rotation, it would be more meaningful. However, the main function of the ACL is limiting the anterior tibial translation. Therefore, we performed this biomechanical study under a 134 N anterior tibial loading condition. Actually, there have been many biomechanical studies under similar loading conditions as our study [13,32,35,37,38]. Li et al. presented in their cadaver study anterior tibial translation and graft forces under 130 N anterior loading at 0, 15, 30, 60, and 90° flexion after conventional SB ACL reconstruction in comparison to an intact knee [13]. Sakane et al. also analyzed in situ forces in the ACL and force distribution to the AM and PL bundle under an anterior load from 22 to 110 N at 0–90° flexion using a cadaver [32]. In their FEM study, Song et al. presented the force and stress distribution within the ACL (AM and PL bundle) under an anterior load (0–134 N) at full extension. Third, we used just one tendon diameter regardless of the knee size. The graft diameter can affect the biomechanics after ACL reconstruction. In some previous studies [39,40], the authors have presented that the ACL graft size affects the degree of impingement on the intercondylar notch and stresses occur within the ACL graft during flexion of the knee using Lachman simulation. Proper determination of the graft size corresponding to the knee size would be included in suture studies [15,41]. Fourth, we were not able to compare anterior translation and graft stress patterns between the two ACL reconstruction techniques and a normal ACL. However, it would have been difficult to conclude whether any ACL reconstruction technique was superior, since the normal ACL stress pattern was not yet known. Fifth, we performed the remnant preservation technique only in SB ACL reconstruction. However, the femoral and tibial tunnel positions were not different from the non-remnant preservation SB ACL reconstruction. In this FEM study, we used only data of the mean tunnel position and the tunnel direction after the ACL reconstruction. Therefore, the remnant preservation technique would not affect the result of this FEM study. Finally, we couldn’t compare the direct effect of only the femoral reaming technique (TT vs. TP or the number of grafts (SB vs. DB), because these two variables were different in each two groups. However, in a previous study, ref. [11] we found that the femoral graft bending angle was significantly different between anatomical DB ACL reconstruction with the TP technique and conventional SB ACL reconstruction with the TT technique and hypothesized that this more acute graft bending angle after the anatomical reconstruction technique would increase the graft stress than the conventional technique. We wanted to evaluate the effect of the anatomical femoral reaming technique (TP) compared to the conventional technique (TT) following our previous study that compared the femoral graft bending angle. So, we performed this FEM study.

## 5. Conclusions

The total graft stress after anatomical DB ACL reconstruction using the TP technique was greater from 0° to 30° flexion and was lesser from 60° to 90° flexion than after conventional SB ACL reconstruction using the TT technique. The total contact stress at the tibial and femoral tunnels was greatest at 0° knee flexion and decreased from 0° to 90° flexion after ACL reconstruction using both techniques. The total contact stresses at the femoral and tibial tunnels after anatomical DB ACL reconstruction were greater than those after conventional SB ACL reconstruction using the TT technique at 0° to 90° knee flexion, respectively. However, the graft and contact stresses of each AM/PL femoral and tibial tunnel after anatomical DB ACL reconstruction were less than those after conventional SB ACL reconstruction.

## Figures and Tables

**Figure 1 jcm-10-01625-f001:**
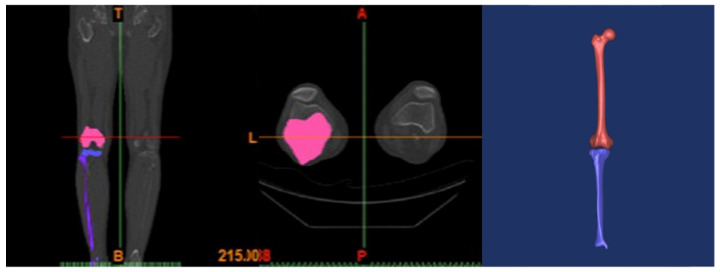
A three-dimensional surface reconstruction of the femur and tibia of a 34-year-old male subject, using Mimics (Materialise Inc., Leuven, Belgium).

**Figure 2 jcm-10-01625-f002:**
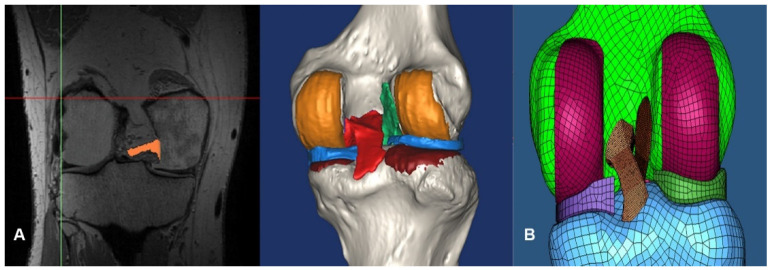
(**A**) A three-dimensional soft-tissue reconstruction based on magnetic resonance imaging (MRI); articular cartilage, both menisci, patellar tendon, and four ligaments (anterior cruciate, posterior cruciate, medial collateral, and lateral collateral ligaments). (**B**) Generation of the finite element model (FEM) using Hypermesh 8.0 (Altair Engineering, Inc., Troy, MI, USA).

**Figure 3 jcm-10-01625-f003:**
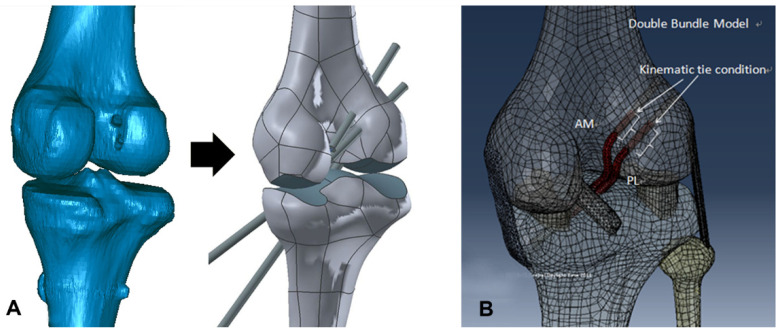
(**A**) Computed tomography (CT) images of each group were used for 3D reconstruction in Mimics. To simulate single-bundle or double-bundle ACL reconstructions in the 0° analytic model, tunnels were reamed at the center of tunnels of the femur and tibia of the validation model using the average femoral tunnel position measured by Bernard’s quadrant method [24] and drilled according to the femoral tunnel direction, estimated by the femoral graft bending angle described by Wang et al. [11] (**B**) The reconstructed ACL was attached to the femoral and tibial tunnels in the model.

**Figure 4 jcm-10-01625-f004:**
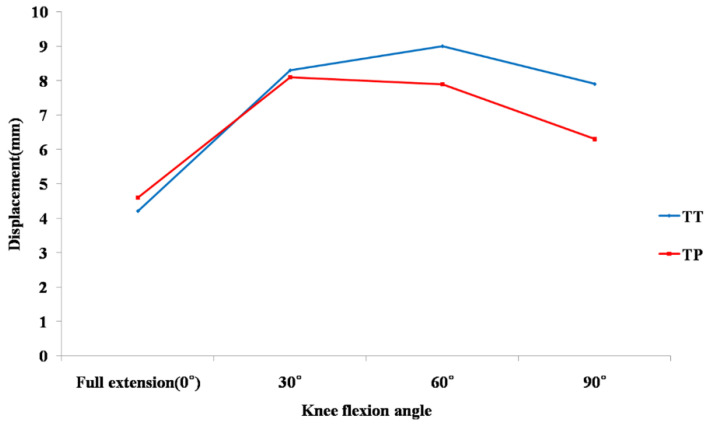
Comparison of anterior tibial translation under a 134 N anterior load between conventional single-bundle and anatomical double-bundle anterior cruciate ligament reconstruction. Abbreviations: TT, transtibial; TP, transportal.

**Figure 5 jcm-10-01625-f005:**
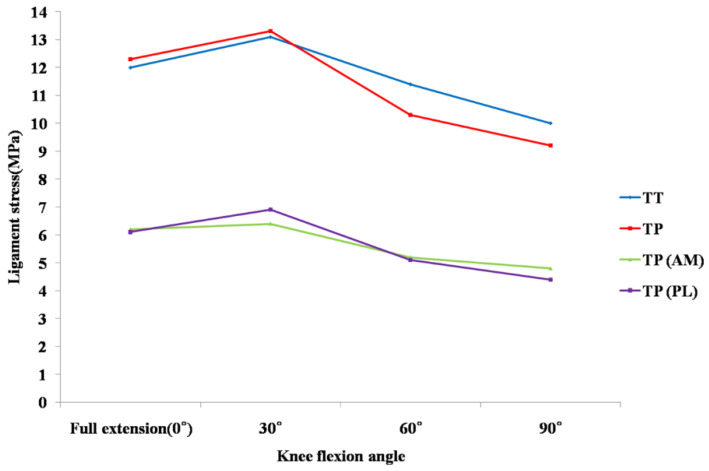
Comparison of graft stress under a 134 N anterior load between conventional single-bundle and anatomical double-bundle anterior cruciate ligament reconstruction. Abbreviations: TT, transtibial; TP, transportal; AM, anteromedial; PL, posterolateral.

**Figure 6 jcm-10-01625-f006:**
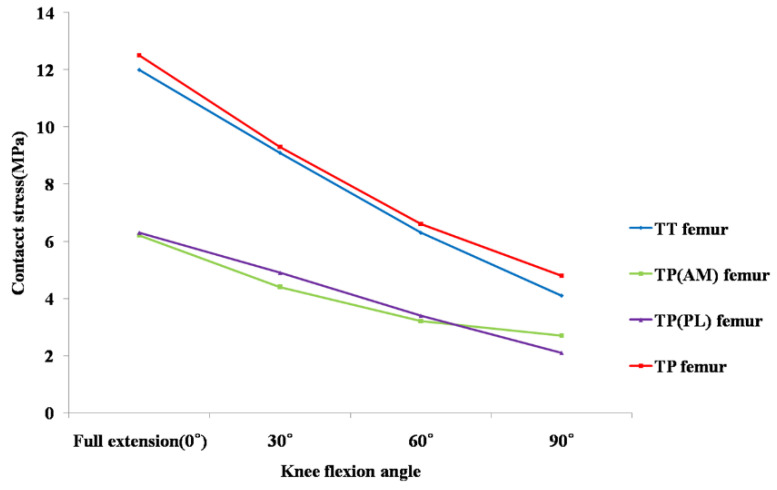
Comparison of contact stress at the femoral tunnel under 134 N anterior tibial load between conventional single-bundle and anatomical double-bundle anterior cruciate ligament reconstruction. Abbreviations: TT, transtibial; TP, transportal; AM, anteromedial; PL, posterolateral.

**Figure 7 jcm-10-01625-f007:**
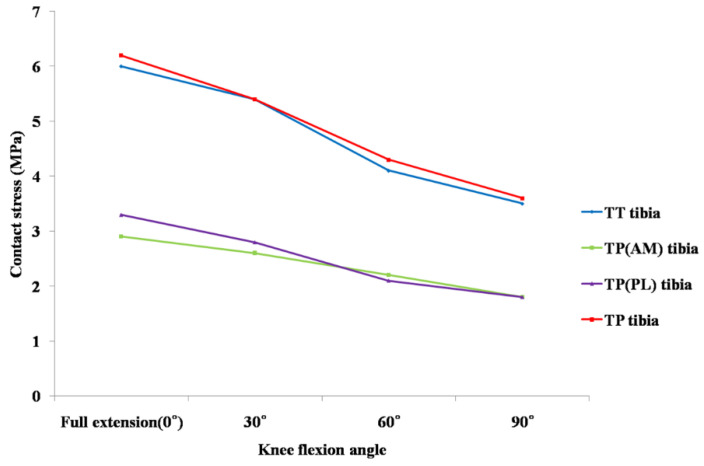
Comparison of contact stress at the tibial tunnel under 134 N anterior tibial load between conventional single-bundle and anatomical double-bundle anterior cruciate ligament reconstruction. Abbreviations: TT, transtibial; TP, transportal; AM, anteromedial; PL, posterolateral.

**Table 1 jcm-10-01625-t001:** Graft stress during knee flexion combined with a 134 N anterior tibial load.^a^

	0°	30°	60°	90°
Transtibial SB ^b^	12.0	13.1	11.4	10.0
Transportal DB ^c^	12.3	13.3	10.3	9.2
AM ^d^	6.2	6.4	5.2	4.8
PL ^e^	6.1	6.9	5.1	4.4

^a^ All data are expressed in MPa. ^b^ single bundle; ^c^ double bundle; ^d^ anteromedial; ^e^ posterolateral.

**Table 2 jcm-10-01625-t002:** Contact stress between femoral tunnel and graft during knee flexion combined with a 134 N anterior load.^a^

	0°	30°	60°	90°
Transtibial SB ^b^				
Femur	12.0	9.1	6.3	4.1
Transportal DB ^c^	12.5	9.3	6.6	4.8
AM ^d^ femur	6.2	4.4	3.2	2.7
PL ^e^ femur	6.3	4.9	3.4	2.1

^a^ All data are expressed in MPa. ^b^ single bundle; ^c^ double bundle; ^d^ anteromedial; ^e^ posterolateral.

**Table 3 jcm-10-01625-t003:** Contact stress between tibial tunnel and graft during knee flexion combined with a 134 N anterior load.^a.^

	0°	30°	60°	90°
Transtibial SB ^b^				
Tibia	6.0	5.4	4.1	3.5
Transportal DB ^c^	6.2	5.4	4.3	3.6
AM ^d^ tibia	2.9	2.6	2.2	1.8
PL ^e^ tibia	3.3	2.8	2.1	1.8

^a^ All data are expressed in MPa. ^b^ single bundle; ^c^ double bundle; ^d^ anteromedial; ^e^ posterolateral.

## Data Availability

Data available in a publicly accessible repository

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
