# Peer review of "Biomechanical Difference between Conventional Transtibial Single-Bundle and Anatomical Transportal Double-Bundle Anterior Cruciate Ligament Reconstruction Using Three-Dimensional Finite Element Model Analysis"

_jcm, 2021, doi:10.3390/jcm10081625_

Round 1
Reviewer 1 Report
While I appreciate the effort of the work, but it requires revision and it still haven’t changed after the first comment. It needs improvement.
Abstract
Basically, well written.
Introduction
Basically, well written.
Material and Methods
Line 78, 90, Figure 1,2
Figure scaling should be uniform in height and width.
Line 113-114, ‘Patients were divided into a conventional TT SB group (20 patients) and an anatomical TP DB group (29 patients). ‘
It is not very convincing that you divided two groups because of the preoperative period. It is important to unified other than what you want to compare. If you want to know the effect of remnant tissue, it is ok. But in my opinion, you should focus on the tunnel position and angle.
Line 129-132, ‘We used both semitendinosus and ~for PL graft..’
How much was the graft diameter in each group respectively? Please explain like ‘AM **mm ± **mm, PL **mm ± **mm’. Every graft can’t be the same size. If you want to discuss about graft tension, it is better to consider the diameter.
Line 163, The mean position of AM and PL ~ .
The exact tunnel position of the tunnels should be stated in the result.
You should explain the position where you planning to be.
Results
Basically, well written and the results are clear and easy to understand.
Discussion
Line 356, Therefore, even~ at the tunnel.
Is this true? DB reconstruction is not for reducing the contact stress.
Reviewer 2 Report
The paper can accept in this form
Reviewer 3 Report
The authors have not addressed my comment sufficiently. A model including varus/valgus forces and rotational forces is required.
Mentioning this as a limitation does not suffice, especially considering the many studies (the authors mentioned) evaluating the ACL under similar loading conditions.
Round 2
Reviewer 3 Report
The authors still have not addressed my points of different loading forces (e.g. rotation or varus/valgus) which are clinically more relevant than flexion and extension.
Author Response
Please see the attachment.

This manuscript is a resubmission of an earlier submission. The following is a list of the peer review reports and author responses from that submission.
Round 1
Reviewer 1 Report
‘Biomechanical Comparison between Conventional Transtibial Single-Bundle and Anatomical Transportal Double-Bundle Anterior Cruciate Ligament Reconstruction using 3-dimensional Finite Element Model Analysis’
This study analyzed the graft stress and contact stress at the tunnel after transtibial SB and transportal DB ACL reconstruction using the post-operative 3-D CT image and concluded total graft and total contact stress after DB were higher than after SB from 0° to 30° and 0° to 90° of knee flexion. There are several points to need to be more explained and revised. While I appreciate the effort of the work, but it requires major revision before ready for publication, in its present form it should be rejected.
Abstract
Basically, well written.
Introduction
Line 63-68, ‘The purpose of this study~ .’
It is an interesting observation to know the graft stress and contact stress at the tunnel after ACL reconstruction. It is not appropriate that to compare the graft tunnel after conventional SB ACL reconstruction and anatomic DB ACL reconstruction. In order to compare differences in one element, other conditions need to be met. It could be worth checking the graft stress and contact stress at the tunnel after anatomical SB ACL reconstruction and anatomic DB ACL reconstruction. When we did the different procedures in ACL reconstruction, ex; transtibial, transportal, outside-in technique, the tunnel position is easily changed. And graft bending angle will change. Also, it is important to know where the tunnels are. You should show the tunnel position and graft angle and compare the two groups. This study design should be reconsidered.
Material and Methods
Line 77-80, Figure 1
Figure scaling should be uniform in height and width. It is easier to visualize if you correct the scale.
The 2 figures in the bottom row are not necessary.
Line 90-94, Figure 2
Figure 2 A, What does it mean in the green color of the MRI? Please explain clearly or turn off the color.
Figure 2 B, Same as Figure 1, figure scaling should be uniform in height and width. Please correct the scale.
Line 113-114, ‘Patients were divided into a conventional TT SB group (20 patients) and an anatomical TP DB group (29 patients). ‘
Please explain how to divide two groups. Please add a description of how you divided it into two groups.
Line 117-120, ‘For SB reconstruction, four-stranded grafts of semitendinosus and gracilis was made and for DB reconstruction six-stranded grafts, which were composed of triple semitendinosus (for AM bundles) and triple gracilis (for PL bundles), were created.’
How much was the graft diameter in each group respectively? Did you take gracilis from all the patients? How big the graft diameter did you aim for and how big the graft diameter was? The Graft diameter must affect the tension applied for the graft.
Line 123, ‘the tibial guide tip was positioned on the tibial footprint (PL bundle center). ‘
The tibial attachment of the ACL is also still a matter of debate. You should explain the landmark clearly.
Line 146, ‘The knee was placed in full extension. ‘
Could all the patients place the knee in full extension 3 days after ACL reconstruction?
Results
Basically, well written and the results are clear and easy to understand.
Discussion
It is not clear how the result of this study will be useful in clinical cases. It is known that anatomical DB ACL-R has better rotational stability, and the difference between TT and TP has been debated so far. What exactly should be changed about these results? Comparing conventional SB ACL reconstruction using TT approach and anatomical DB ACL reconstruction using TP approach does not lead to the answer.
Reviewer 2 Report
Dear authors:
It has been a pleasure to review your paper “Biomechanical Comparison between Conventional Transtibial Single-Bundle and Anatomical Transportal Double-Bundle Anterior Cruciate Ligament Reconstruction using 3-dimensional Finite Element Model Analysis” and you only need to change a few errors that it’s necessary to accept it. You can see below the recommendation
Abstract
Please can you include the long name before doing the abbreviature?
Can you include anything of the method in this section?
Reviewer 3 Report
The authors evaluate the stress on different ACL reconstruction methods during flexion and extention. The paper could be improved if the authors were to evaluated varus/valgus forces and rotational forces as these are paramount in the tear of the "original" ACL.
Additionally, often a combination of F/E, varus/valgus and rotations puts the highest strains on the graft.
